# Common Bean Productivity and Micronutrients in the Soil–Plant System under Residual Applications of Composted Sewage Sludge

**DOI:** 10.3390/plants12112153

**Published:** 2023-05-30

**Authors:** Gabriela Souza de Oliveira, Arshad Jalal, Adrielle Rodrigues Prates, Marcelo Carvalho Minhoto Teixeira Filho, Rodrigo Silva Alves, Luana Corrêa Silva, Raimunda Eliane Nascimento do Nascimento, Philippe Solano Toledo Silva, Orivaldo Arf, Fernando Shintate Galindo, Fernando Carvalho Oliveira, Cassio Hamilton Abreu-Junior, Arun Dilipkumar Jani, Gian Franco Capra, Thiago Assis Rodrigues Nogueira

**Affiliations:** 1Department of Plant Protection, Rural Engineering and Soils, São Paulo State University, Av. Brasil n◦ 56, Ilha Solteira 15385-000, SP, Brazilarshad.jalal@unesp.br (A.J.); adrielle.prates@unesp.br (A.R.P.); mcm.teixeira-filho@unesp.br (M.C.M.T.F.); rodrigo.s.alves@unesp.br (R.S.A.); 2Department of Agricultural Sciences, School of Agricultural and Veterinarian Sciences, São Paulo State University, Via de Prof Access Paulo Donato Castellane, s/n, Jaboticabal 14884-900, SP, Brazil; luana-correa.silva@unesp.br (L.C.S.); philippe.toledo@unesp.br (P.S.T.S.); 3Department of Plant Technology, Food Technology and Partner Economics, São Paulo State University, Av. Brazil Sul n◦ 56, Ilha Solteira 15385-000, SP, Brazil; o.arf@unesp.br; 4School of Agricultural and Technological Sciences, Sao Paulo State University, Rod. Captain João Ribeiro de Barros km 651, Dracena 17900-000, SP, Brazil; fernando.galindo@unesp.br; 5Tera Ambiental Ltda., Estrada Municipal do Varjão n◦ 4.520, Jundiaí 13212-590, SP, Brazil; fernando.carvalho@teraambiental.com.br; 6Center for Nuclear Energy in Agriculture, University of Sao Paulo, Av. Centenário n◦ 303, Piracicaba 13416-000, SP, Brazil; cahabreu@cena.usp.br; 7Department of Biology and Chemistry, California State University, Monterey Bay, Seaside, CA 93955, USA; ajani@csumb.edu; 8Dipartimento di Architettura, Design e Urbanistica, Università degli Studi di Sassari, Polo Bionaturalistico, Via Piandanna n◦ 4, 07100 Sassari, Italy; pedolnu@uniss.it; 9Desertification Research Centre, Università degli Studi di Sassari, Viale Italia n◦ 39, 07100 Sassari, Italy

**Keywords:** *Phaseollus vulgaris* L., food security, organic fertilizers, plant nutrition, urban waste

## Abstract

Composted sewage sludge (CSS) is an organic fertilizer that can be used as a source of micronutrients in agriculture. However, there are few studies with CSS to supply micronutrients for the bean crop. We aimed to evaluate micronutrient concentrations in the soil and their effects on nutrition, extraction, export, and grain yield in response to CSS residual application. The experiment was carried out in the field at Selvíria-MS, Brazil. The common bean cv. BRS Estilo was cultivated in two agricultural years (2017/18 and 2018/19). The experiment was designed in randomized blocks with four replications. Six different treatments were compared: (*i*) four increasing CSS rates, i.e., CSS5.0 (5.0 t ha^−1^ of applied CSS, wet basis), CSS7.5, CSS10.0, CSS12.5; (*ii*) a conventional mineral fertilizer (CF); (*iii*) a control (CT) without CSS and CF application. The available levels of B, Cu, Fe, Mn, and Zn were evaluated in soil samples collected in the 0–0.2 and 0.2–0.4 m soil surface horizons. The concentration, extraction, and export of micronutrients in the leaf and productivity of common beans were evaluated. The concentration of Cu, Fe, and Mn ranged from medium to high in soil. The available levels of B and Zn in the soil increased with the residual rates of CSS, which were statistically not different from the treatments with CF. The nutritional status of the common bean remained adequate. The common bean showed a higher requirement for micronutrients in the second year. The leaf concentration of B and Zn increased in the CSS7.5 and CSS10.0 treatments. There was a greater extraction of micronutrients in the second year. Productivity was not influenced by the treatments; however, it was higher than the Brazilian national average. Micronutrients exported to grains varied between growing years but were not influenced by treatments. We conclude that CSS can be used as an alternative source of micronutrients for common beans grown in winter.

## 1. Introduction

Brazil is among the world’s largest common bean (*Phaseolus vulgaris* L.) producers [1]. The estimated production was around 1.43 million tons from the planted area of 1.38 million hectares in 2021/2022, reflecting an increase in productivity [2]. Beans are produced and imported in many countries, as they are one of the main sources of protein, rich in macro- and micronutrients, such as K, Mg, Fe, and Zn, among other benefits [3,4].

Brazil produces three crops of common beans annually, which vary between regions. The first is the rainy season, where common beans are sown between August and November. The second is the dry season, sown from November to March. The third is winter or irrigated from April to July [5,6]. The production of winter beans requires more technology than crops in appropriate weather conditions, and the use of good-quality seeds, fertilizers, correctives, and pesticides makes it possible to obtain three- to five-times higher production than sowing times [6,7].

Common bean cultivation with sprinkler center pivot irrigation is the most common method farmers use in the Brazilian Cerrado region of the Midwest and Southeast states of Brazil [5]. It is necessary to overcome phytosanitary problems, such as pests and diseases, as well as climate changes, such as water availability and temperature. In addition, soils in this region are under constant weathering, which reduces quality, causing high acidity, less organic matter with low cation exchange capacity (CEC) and anion exchange capacity (AEC), and deficiency in macro- and micronutrients [8].

The common bean has high requirements in terms of soil fertility due to its short cycle and superficial root system [9]. The common bean requires micronutrients in small amounts, but these are essential for obtaining productivity gains [10,11]. Thus, adequate nutritional management through the correction of acidity of tropical soils is essential to achieve high gain in the final production of common beans [12,13,14]. However, producers face an annual increase in the prices of mineral fertilizers, especially in countries with a high dependence on the imports of fertilizers, such as Brazil [15]. Thus, it becomes necessary to search for alternative sources of nutrients to complement mineral fertilization and replace this dependence, causing global food insecurity.

The recycling of organic waste in agricultural soils, specifically sewage sludge (SS), is a promising alternative due to its sustainable final disposal by-product that has been produced on a large scale in urban centers [16,17]. Sewage sludge comes from the treatment of effluents and contains high levels of organic matter and plant nutrients that can be used in agriculture as a fertilizer [18,19]. Sewage sludge composting is a strategy to reduce the volume of SS and make it safer for application in agricultural areas for better management [20]. The proper treatment of sewage sludge via composting is called composted sewage sludge (CSS). Suppose CSS meets the criteria established by the Ministry of Agriculture, Livestock and Supply of Brazil in a Normative Instruction No. 61 of 8 July 2020, it can be registered as a Class B organic fertilizer [21]. Recently, there has been evidence of the agronomic viability of using CSS as an organic fertilizer, especially for supplying P, N, and micronutrients [22,23,24,25,26,27].

Based on the above, we hypothesized that the residual effect of CSS applications would provide micronutrients to the soil in sufficient quantities to maintain the plants’ nutritional balance, guarantee the productivity of winter beans, and reduce the application of mineral fertilizers. Thus, our study evaluated the available levels of micronutrients in the soil and their effects on nutrition, extraction, and export associated with the productivity of common beans grown in the soil of the Cerrado region. According to the case or necessity, it is made internally with external consultations.

## 2. Results

### 2.1. Soil Micronutrients

Cationic micronutrients, such as Cu, Fe, Mn, and Zn, remained higher in available concentrations in the soil when compared to the concentration of anionic micronutrients such as B (Figure 1). The concentrations of Cu and Zn in the soil were higher at a depth of 0–0.20 m. Such results were also observed in the two agricultural years evaluated (Figure 1b,e). It was also noted that the availability of B, Mn, and Zn increased in the soil in the second agricultural year.

In the case of a depth of 0.20–0.40 m, only concentrations of Mn and Zn varied between the applied treatments, while concentrations of Cu, Fe, Mn, and Zn varied in the function of agricultural years (Figure 2). Residual rates of CSS above 7.5, 10.0, and 12.5 Mg ha^−1^ resulted in higher concentrations of B, Cu, Fe, and Zn. In addition, the highest available soil Mn concentration was verified in CT and with CF. The same results were observed at two depths (0.0–0.20 and 0.20–0.40 m). However, the highest Mn concentration was observed in the CTs in the 0.20–0.40 m layer. Still, at a depth of 0.20–0.40 m, it was found that the levels of B and Zn in the soil were similar in both agricultural years. However, it was possible to notice greater availability of Cu, Fe, and Mn at a soil depth of 0.20–0.40 m for the 2017/18 agricultural year.

The limits established by Boaretto et al. [11] for the interpretation of concentrations of micronutrients in soil for annual crops are described in Figure 1. Thus, it was noted that the soil B concentration was increased in the second agricultural year, changing from a low concentration (<0.2 mg dm^−3^) for medium (0.2–0.60 mg dm^−3^), and Cu, Fe, and Mn concentrations showed high levels (Cu: >0.8; Fe: >12, and Mn: >5.0 mg dm^−3^) in two agricultural years. Except CT and CF in 2017/18, it was observed that the available soil Zn concentration was considered high (<1.2 mg dm^−3^) in both agricultural years. These results proved that the residual CSS rates increased soil fertility by increasing concentrations of B, Cu, Fe, Mn, and Zn.

Considering the residual effect of CSS rates at a depth of 0–0.20 m, the available B and Zn concentrations were linearly increased in the first and second agricultural year, respectively (Figure 3). At a depth of 0.20–0.40 m, there was a linear increase in soil Zn availability in response to the residual effect of applied CSS rates (Figure 4). However, although soil Zn concentrations were increased in both years of study, the response was more expressive in the first agricultural year as compared to the second year.

### 2.2. Plant

#### 2.2.1. Leaf Micronutrient Concentrations

In the 2017/18 agricultural year, there was a difference between treatments only for leaf Zn concentrations. The treatment with CF and highest rates of CSS were observed with a higher leaf Zn concentration. In addition, there was variation in the leaf concentration of all micronutrients under the studied treatments in the second year of bean cultivation (2018/19) (Figure 5). In general, the residual rates of CSS provided leaf micronutrient concentrations similar to the treatment with CF. The CT without mineral and organic fertilizer but with application of limestone and agricultural gypsum before the installation of the experiment also provided similar results to CSS and CF.

For all the treatments tested in the first year of cultivation, it was noted that the leaf B concentration exceeded the proposed limit (26 mg kg^−1^ of B) for common bean crops (Figure 5). However, no symptoms of visual toxicity were observed in the plants in field conditions. In the second year, it was noted that the plants in the CSS10.0 treatment were within the appropriate range of leaf B levels. In addition, the plants had a high leaf B concentration with the other treatments with CF or CSS. The leaf concentrations of Fe, Cu, Mn, and Zn remained within the range considered adequate for the common bean crop (Figure 5).

It was possible to notice a difference in micronutrient concentrations in bean leaves in response to the two agricultural years (Figure 5). Thus, it was also verified that the treatments with CF and residual effect of the first year of CSS were observed with higher leaf B concentrations of common beans. The leaf Cu concentration was also higher in the first year of cultivation for the treatments with CT, CF, and with CSS5.0. There was no variation in Cu concentration for the other treatments as a function of the year of study. Leaf Fe concentrations were higher only in the first year of common bean cultivation under the CF treatment. The leaf Mn concentration of common beans was higher only in the CSS7.5 treatment in the first and second year of plant cultivation. The leaf Zn concentration was higher in the plants cultivated in the first year of the study within CT and CSS7.5 treatments.

Evaluating the residual effect of CSS rates on the leaf micronutrient concentrations, a linear increase in Zn concentrations was observed with the application of CSS rates in the first agricultural year (2017/18) (Figure 6a). In addition, leaf B and Cu concentrations were not influenced by the residual effect of CSS rates in the first year (Figure 6b,c). In the second agricultural year (2018/19), the residual effect of CSS rates linearly increased the leaf B concentration (Figure 6a). Leaf Cu concentration was set to a quadratic adjustment, being observed that the rate of 5.9 t ha^−1^ of CSS (wet basis) provided the highest Cu concentration (17.70 mg kg^−1^) in common bean leaves (Figure 6b).

#### 2.2.2. Extraction and Export of Micronutrients

Extraction refers to the accumulation of nutrients from the shoot of the plant, without accounting for the grains. Export is the accumulation of nutrients only in the grains (*vide infra*).

Micronutrient extraction for winter beans was significantly influenced by the treatments (Figure 7). The treatments with CF showed the highest B extractions in both agricultural years. The concentration of B extraction ranged from 26.64 g ha^−1^ in the 2017/18 agricultural year to 50.02 g ha^−1^ of B in the 2018/19 agricultural year (Figure 7a). It was noted that CSS applications increased the shoot B content in common bean plants as compared to the control treatment in both agricultural years, with average B contents ranging from 14.17 to 44.28 g ha^−1^. Hence, the average concentration of extracted B by plants was increased by 5.25 g ha^−1^ in the first year and 20.89 g ha^−1^ in the second year, which probably came from the correction of acidity through limestone application in the initial phase of the experiment, contributing to higher physiological efficiency of the plant in the extraction from one year to the next.

The extraction of both Fe and Mn was higher in the CF treatments in both agricultural years (Figure 7c,d). The Fe extraction was increased by 370.92 and 632.30 g ha^−1^, while Mn extraction was increased by 130.50 and 161.32 g ha^−1^ in the first and second year (Figure 7c,d).

There were different responses of the extraction of Cu and Zn in both years of bean cultivation (Figure 7b,e). The highest Cu extraction was observed in the first year of CSS5.0 treatment. The highest amount of Zn was extracted with all treatments except CT in the second year of cultivation. However, in both studied years, the treatments with CF were observed with the highest extractions of 9.33 g ha^−1^ of Cu in the first year and 47.02 g ha^−1^ of Zn in the second year. The lowest extraction was observed in CT in the first year of cultivation, with average concentrations of 4.24 g ha^−1^ of Cu (Figure 7b) and 23.17 g ha^−1^ of Zn (Figure 7e). The highest values of Zn extraction were noted with CF, which was statistically similar with residual CSS rates in the 2018/19 agriculture year (Figure 7e).

The order of micronutrients extracted by the bean shoot was Fe > Mn > Zn > B > Cu. The adequate availability of the basic nutritional needs regulates the ability of bean plants to extract nutrients. The extractions of B, Fe, and Mn were increased linearly in the first agricultural year as a function of the application of CSS rates (Figure 8), demonstrating that the application of organic fertilizer increased absorption of these nutrients by plants. The CSS rates did not influence the extraction of B, Fe, and Mn in the second year, with average concentrations of 33.99, 354.38, and 74.82 g ha ^1^, respectively.

The export of micronutrients carried out by the bean plant (grains) (Figure 9) indicated that the relationship between treatments and export was significant for B and Cu. The highest exported content of B was recorded in the first year and Cu in the second year of common bean cultivation. There were no significant differences among the CT and CSS rates and CF for the means of B content. The means of Cu in the first year were not different, while there was variation for the export of Cu in the second year, where the highest average (83.86 g ha^−1^) was observed in CSS12.5 treatments, i.e., those with the highest applied CSS rate.

The exported quantities of Fe did not change in both of the agricultural years. The export of B and Zn was significant with the treatments. The export of B showed the highest values in the first year and Zn in the second year. The highest values of Cu were obtained in the second year of cultivation, except the rate of 7.5 t ha^−1^ of CSS, which did not differ from the other treatments. The exports of Mn were different in all treatments, while the highest Mn export was noted in the first agricultural year.

The order of micronutrients exported in the winter bean crop had a different behavior in both years, with a decreasing order in 2017/18; Fe > Zn > Mn > B > Cu and in 2018/19; Fe > Zn > Cu > B > Mn. However, most of the micronutrients were exported more by plants treated with CSS application, as compared to CT and CF treatments in the second year.

Comparing the amounts of micronutrients in the grains and shoots, regardless of the year studied, the shoots accumulated more Fe and Mn, while grains were characterized by greater accumulations of Fe and Zn. It was noted that common beans exported more Cu in the second agricultural year that ranked this nutrient from the least exported to the third most exported nutrient.

The amount of the (grain) export of Fe, Mn, and Zn was linearly increased with increasing CSS rates (Figure 10). However, the increase occurred only in the first year of crop cultivation. In the second year, there was no difference in relation to the applied CSS rates and the amount of micronutrients exported. Thus, these results show the efficiency of CSS in supplying micronutrients to plants, contributing to increased crop productivity in low-consumption agricultural systems.

#### 2.2.3. Grain Productivity

The grain productivity of winter common beans was not influenced by treatments in both agricultural years (Figure 11). The productivity was not statistically different with all CSS and CF treatments.

The results of this research show that the residual effect of the application of CSS rates can be considered a viable alternative in the cultivation of winter beans. In the first agricultural year, grain yield was linearly increased in response to the residual effect of CSS application (Figure 12). In the second year of bean cultivation, there was no influence of the residual rates of CSS on plant productivity, with an average value of 2594 kg ha^−1^ being observed.

## 3. Discussion

The soil cationic micronutrients, such as Cu, Fe, Mn, and Zn, remained in higher concentrations when compared to the anionic B. This behavior is normally observed in the soils with intense weathering, such as the Oxisol investigated in this study (pH-CaCl_2_: 4.5 ± 0.06; Table 1), after residual accumulation of CSS. The increase in soil organic matter, due to CSS application, a by-product naturally enriched in OM (first year: 308.7 ± 10.0 g kg^−1^; second year: 255.0 ± 7.37 g kg^−1^; Table 2), influences the increase in B levels, which is strongly influenced by SOM content, thus becoming very dynamic along the profile due to soil leaching [30], naturally featuring highly weathered Oxisols, as in the present study (Figure 1).

The concentrations of B and Zn were increased at a soil depth of 0.0–0.20 m, while all the studied micronutrients were increased at a depth of 0.20–0.40 m, when compared with the initial contents obtained in the soil (Table 1). This fact demonstrated the ability of organic fertilizer to supply these micronutrients to the soil, except soil Cu concentration, which was reduced in the second year. Among the studied micronutrients, soil Zn concentration was increased at both of the evaluated depths, as also occurred in a study by Silva et al. [26,27], who evaluated the changes in the chemical properties of the soil after CSS application in sugarcane cultivation in Cerrado soil. Several other studies reported that residual CSS improves soil fertility in terms of the supply of micronutrients, such as B, Cu, Fe, Mn, and Zn, in legume cultivation [23,25].

Leaf B concentration is an important factor that remained above the appropriate levels, unlike the other micronutrients that remained within adequate ranges in the leaf; however, no symptoms of toxicity were observed in beans (Figure 5). A study by Prates et al. [24] reported that the leaf B concentration was increased with residual rates of CSS but still no toxicity was seen in plants with B concentrations above the recommended values in the literature [28]. The improvement in biochemical and physiological processes might help the plants to improve the tolerance mechanisms in response to the excess of these micronutrients in soil [34]. The present study indicated that the leaf micronutrient concentrations were increased with CSS. However, there was no increase in the leaf micronutrient concentrations in relation to the application of CF and the CT. The same behavior was observed by Bertolazi et al. [35] and Prates et al. [24] in their studies with eucalyptus and soybean crops, respectively. In all cases, soil Zn concentration had a significant increase with increasing CSS rates, which demonstrated that the absorption of micronutrients is related to soil–plant interactions and also governing the competitive/inhibitory processes with other micronutrients [24].

Synthetic and mineral fertilizers are the most common sources of micronutrients. A wide range of these fertilizers are commercially available, so it is very easy to find a suitable product to meet the needs of plants. However, these fertilizers are very expensive and imported. In addition, for the principles of sustainable agriculture and environmental protection, farmers are paying more attention to the possibility of using organic fertilizers, which are in line with consumer expectations as well as reducing the exploitation of mineral sources and lowering production costs [36].

Organic fertilizers guarantee a gradual release of nutrients, maintaining their positive balance for a long period of time. Jakubus and Bakinowska [22] indicated that the systematic application of organic fertilizers in the long term could not only result in an increase in soil organic matter but also increase the available amounts of micronutrients to plants, as presented in the current experiment (Figure 6 and Figure 7). Thomas et al. [37] listed the positive aspects related to the use of recycled organic waste as agricultural fertilizers. In this sense, an interesting alternative to commonly used mineral fertilizers can be provided by compost prepared from sewage sludge, as proved in the present experiment.

The increase in concentrations of micronutrients in the shoots of common beans evidenced the supply of B through the compost applications (Figure 5); initially, the soil had average levels in the experimental area (Table 1). It is known that the accumulation of micronutrients by plants is strictly related to their available contents in soil and the applied correctives [36]. Furthermore, B plays a key role in plant metabolism, including cell wall formation and stability, which maintains membrane integrity, sugar and energy activation, pollination, pod fixation, and increased grain yield per plant, resulting in higher productivity [38].

In general, the values obtained in the extraction of micronutrients in the present study were lower than those found in the previous study (Figure 7). Perez et al. [39] evaluated different rates and times of nitrogen fertilization via mineral fertilizer (60 kg ha^−1^ pre-sowing; 60 kg ha^−1^ at V4 stage; 60 kg ha^−1^ at pre-sowing + 60 kg ha^−1^ in coverage) in bean crops in a no-tillage system and observed that mineral fertilizer increased micronutrient exportation to the plants. In addition, the higher macronutrient application fulfills greater demand of plants for micronutrients, especially the cultivars that can express high productive potential [40,41].

The compounds prepared from sewage sludge are characterized by higher Fe and lower Mn concentrations [36]. These results are similar to those observed by Prates et al. [24]. These authors suggest that application of CSS promotes an adequate concentration of micronutrients, avoiding deficiencies while reducing toxicity problems in Cerrado soil (Figure 8 and Figure 9).

The results obtained in the first agricultural year corroborated the work of Fageria et al. [42] and Fernandes et al. [40], who evaluated the export of micronutrients in common beans with rates of mineral fertilizers. It was noted that crops exported more Cu in the second agricultural year, ranking the least exported nutrient by bean plant to be the third most exported (Figure 9). Fageria et al. [42] evaluated three rates of limestone (0, 12, and 24 Mg ha^−1^), and seven rates of B (0, 2, 4, 8, 12, 16, and 24 kg ha^−1^) in common beans reported the lowest Cu exported values in both shoots and grains.

Silva et al. [43] evaluated the effect of sewage sludge application on shoot biomass, productivity, and soil and plant nutrient concentration in three successive maize crops. These authors obtained similar results to the present study regarding the availability of micronutrients. Rodrigues et al. [44] also highlighted that the availability of micronutrients increases the use efficiency of biosolids in agriculture. Furthermore, Prates et al. [23], evaluating the effect of applying CSS in low-fertility soils in the Cerrado region, concluded that CSS could be an excellent alternative to mineral fertilization as a source of micronutrients for soybean cultivation.

Common bean productivity exceeded the Brazilian average in both studied years, with an average of 1268 kg ha^−1^ in the 2017/18 crop and 1474 kg ha^−1^ in the 2018/19 crop (Figure 11 and Figure 12). It was observed that productivity in the current study was 101 and 82% higher than the average Brazilian productivity in the first and second common bean harvest (Figure 11). Using this general average of treatments with CSS application, it was also noted that the present productivity surpassed the other studies as well, such as that by Sandrini et al. [45], whose common bean productivity was 2329 kg ha^−1^ as a function of nitrogen application rates. In addition, Prates et al. [23] reported a linear increase in soybean productivity with the application of CSS rates in the first year of soybean cultivation. As verified in the present study for winter beans, some previous studies have already reported that CSS is capable of being an alternative source of micronutrients, increasing the productivity of several crops (e.g., sugarcane, soybean, and maize) while completely eliminating the use of mineral fertilizers in some situations [23,24,26]. However, it is essential to monitor micronutrient levels in the soil–plant system, always aiming to maintain the levels of these nutrients above the critical levels established for each type of soil and agricultural region. Still, it is important to assess the availability of macronutrients in the soil, favoring the nutritional balance for all nutrients and maintaining crop productivity.

## 4. Materials and Methods

### 4.1. Site Description

The current study was carried out under field conditions in agricultural years of 2017/18 and 2018/19 at Selvíria-MS, Brazil (Figure 13). The region has an annual average rainfall of 1370 mm, temperature of 24.5 °C, and relative humidity of 75%; thus, it was characterized as rainy summers and dry winters by the Aw climate according to Köppen classification [46]. The monthly data regarding maximum and minimum temperatures, relative humidity, and rainfall were collected throughout the experimental period (Figure 14).

The soil of the experimental area was classified as a Rhodic Hapludox [47], presenting physical and chemical attributes, as described in Table 1.

Each experimental unit consisted of seven rows of bean crops (spaced 0.45 m apart) of 10 m in length, totaling 31.5 m^2^ per plot and 1260 m^2^ of total experimental area. The useful area of the plot for the evaluations consisted of the three central lines, 2.5 m from each end being eliminated as a border, totaling approximately 7.0 m^2^.

### 4.2. Experimental Design and Treatments

A randomized block design with split plots was adopted with six main and two secondary treatments in four replications. The six main treatments consisted of: (*i*) four rates of composted sewage sludge—CSS (5.0, 7.5, 10.0, and 12.5 Mg ha^−1^, wet basis; CSS5.5, CSS7.5, CSS10.0, CSS12.5, hereafter); (*ii*) a control treatment (CT, without application of CSS or mineral fertilizers); and (*iii*) a treatment with conventional mineral fertilizer (CF). The two agricultural years (2017/18 and 2018/19) of common bean growing were used as a secondary treatment. The bean cultivation, treatments, experiment were kept the same for both years, aiming to compare the response to the residual effect of CSS applications.

### 4.3. Obtaining and Characterizing Sewage Sludge Compost

The CSS was obtained from the composting of sewage sludge generated at the wastewater treatment plant in the municipality of Jundiaí, SP, Brazil. The process consists of turning over and drying the material at the same time as using a forced aeration system for three consecutive months. The chemical and microbiological characterization of CSS was carried out (Table 2), as recommended by Conama Resolution No. 498 [48].

### 4.4. Installation and Development of the Study

The initial preparation of the area was carried out via scarification in September 2017. Based on the results of the soil fertility assessment, the area was applied with 2.2 Mg ha^−1^ of limestone, aiming to raise base saturation to 70%, and then 1.8 Mg ha^−1^ of agriculture gypsum was applied on a surface without incorporation [49].

The common bean (*Phaseolus vulgaris* L.), cv. BRS Estilo, was used as a test plant. Bean plants were cultivated in the winter of 2017/18 and 2018/19 growing years. A desiccation of the vegetation cover was carried out with the application of herbicides Verdict R (37.41 g ha^−1^ active ingredient-ai) and Gramoxone (300 g ha^−1^ ai). The application of CSS was carried out in rice cultivation with two consecutive applications prior to common bean in the experimental area. The CSS was manually distributed over the total area within each experimental plot.

Before sowing the crop, the seeds were treated using fungicides thiophanate methyl + pyraclostrobin (5 g + 45 g ai per 100 kg of seed) and insecticide fipronil (50 g ai per 100 kg of seed). Bean seeds were primed with solution of cobalt and molybdenum-CoMo (39.21 mL) and *Rhizobium tropici* (80 g of inoculant for each 50 kg of seed) together with a 60 mL sucrose solution to stimulate nodulation.

Aiming to evaluate micronutrients in soil and in plant, it was decided to perform mineral fertilization with CSS to supply macronutrients. For this purpose, chemical analysis of the soil and recommendations in Bulletin-100 [49] were taken into account. A rate of 400 kg ha^−1^ of NPK fertilizer (04–20–20), corresponding to 16 kg ha^−1^ of N, 80 kg ha^−1^ of P_2_O_5,_ and 80 kg ha^−1^ of K_2_O, was applied in both growing seasons of common beans. The plots referring to the treatment with conventional mineral fertilization were applied with 1.0 kg ha^−1^ of boron using boric acid (H_3_BO_3_) in the first year (2017/18) while soil B level was adequate in the agricultural year (2018/19).

The cover fertilization in common bean crop was carried out according to the recommendations of Ambrosano et al. [50]. A total of 90 kg ha^−1^ of N (via urea) was applied to the plots that received conventional mineral fertilization. This rate (90 kg ha^−1^) was split, and we applied 50.0 and 40.0 kg ha^−1^ of N at 15 and 30 days after sowing (DAS), respectively. In addition, treatments that received CSS were applied with 50 kg ha^−1^ of N (in the form of urea) at 15 DAS. The distribution of mineral fertilizers was carried out on the soil surface without incorporation approximately 0.08 m from the sowing lines in order to avoid contact between fertilizer and plants. After the applications, the area was irrigated using a sprinkler (central pivot) with a 14 mm blade.

### 4.5. Soil and Plant Analysis

#### 4.5.1. Available Levels of Micronutrients in the Soil

Soil samples were collected from 0–0.20 to 0.20–0.40 m deep within the useful area of each plot at the end of the crop cycle. Five random subsamples were collected per plot to compose a composite sample. These samples were taken with the aid of a mechanical sampler. Afterwards, the samples were air-dried, crushed, and passed through a sieve with a 2 mm mesh opening, packed in identified polyethylene bags, and stored in a dry chamber until the time of analysis.

The available concentrations of Cu, Fe, Mn, and Zn in the soil were obtained using a DTPA chemical extractor at pH 7.3 [51], quantified via an inductively coupled plasma atomic emission spectrometer (ICP-AES), Model Varian Vista-MPX, Varian, CA, USA, with respective wavelengths in nm: Cu = 324.7; Fe = 259.9; Mn = 257.6, and Zn = 206.2. The detection limit for the analyzed elements was established by reading blank sample (washed sand) seven times and calculated using the formula LD = 3.s, where “s” is the standard deviation. The concentration of B in the soil was evaluated by means of extraction with barium chloride and heating with a microwave oven and quantified in a UV-VIS spectrophotometer (Model Varian Cary-50, Varian, Victoria, Australia) at 420 nm [51].

#### 4.5.2. Nutritional Parameters of Bean Culture

The nutritional status was evaluated during the R6 stage (flowering) of common bean, grown in the 2017/18 and 2018/19 cropping seasons. The third leaf with petiole was randomly collected from 10 plants per plot, as described by Ambrosano et al. [28]. After being washed, the samples of each plot in the field were dried in an oven with forced air circulation at a temperature of 65 °C for 72 h and crushed in a Willey-type mill. This same material was submitted to wet digestion with nitric acid (HNO_3_) and perchloric acid (HClO_4_) [52] for the determination of Cu, Fe, Mn, and Zn concentrations and dry digestion (muffle incineration) for the determination of B.

For grains and shoots, 10 plants were collected, at the physiological maturation stage, to perform the extraction of the aerial part of the bean crop, multiplied by the nutrient concentration in the dry matter of the aerial part and extrapolated based on the plant population. After measuring grain yield, the grains were used to obtain the accumulation of exported nutrients. The samples were then submitted to the same treatments previously exposed for leaves (*vide supra*) to micronutrient concentration determination.

#### 4.5.3. Export of Micronutrients and Grain Yield

Export refers to the accumulation of nutrients in the grains, which were removed from the system by the harvest. The calculation of exports was carried out through the productivity of the grain per hectare in dry matter by the concentration of nutrients. In particular, to estimate grain yield, all plants in the useful area were harvested and manually threshed to avoid losses, weighed, and calculations were performed with extrapolation to kg ha^−1^ and corrected for 13% moisture content [53]. Samples of these grains were taken for the analysis of micronutrient concentrations and exported amounts of these nutrients per area.

### 4.6. Statistical Analysis

All data were initially tested for normality using Shapiro and Wilk [54] test and Levene’s homoscedasticity test (*p* ≤ 0.05), which showed the normal distribution (W ≥ 0.90) of the data. The results obtained were subjected to analysis of variance, applying the Tukey test (*p* ≤ 0.05) to compare main treatments (CSS rates, control, and conventional fertilization) and secondary treatments (growing years 2017/18 and 2018/19). Additionally, polynomial regression was performed for the effects of CSS rates, using control plot as zero rate. Statistical analysis was performed using the R studio [55] 2022.02.0+443 software.

## 5. Conclusions

Conventional mineral fertilization and residual application of CSS rates provided similar results for all evaluated variables (yield, extraction, and export of micronutrient content in leaf and soil) in the present study. By comparing the agricultural years, it was possible to notice variation within each production cycle. However, there is still no pattern between the analyzed variables, requiring long-term studies. The concentrations of micronutrients in the leaves and in soil were observed within adequate limits. Therefore, it is possible to notice that the residual effect of CSS application was able to supply the nutrients in amounts necessary for a better development of winter bean cultivated in the Cerrado region. Thus, we concluded that CSS can be used as an organic fertilizer to supplement micronutrients, in particular to increase the levels of B and Zn in soil–plant systems. Future studies will focus on the behavior of investigated micronutrients into the soil environment too, both in total and available form, along the two years of experimentation.

## Figures and Tables

**Figure 1 plants-12-02153-f001:**
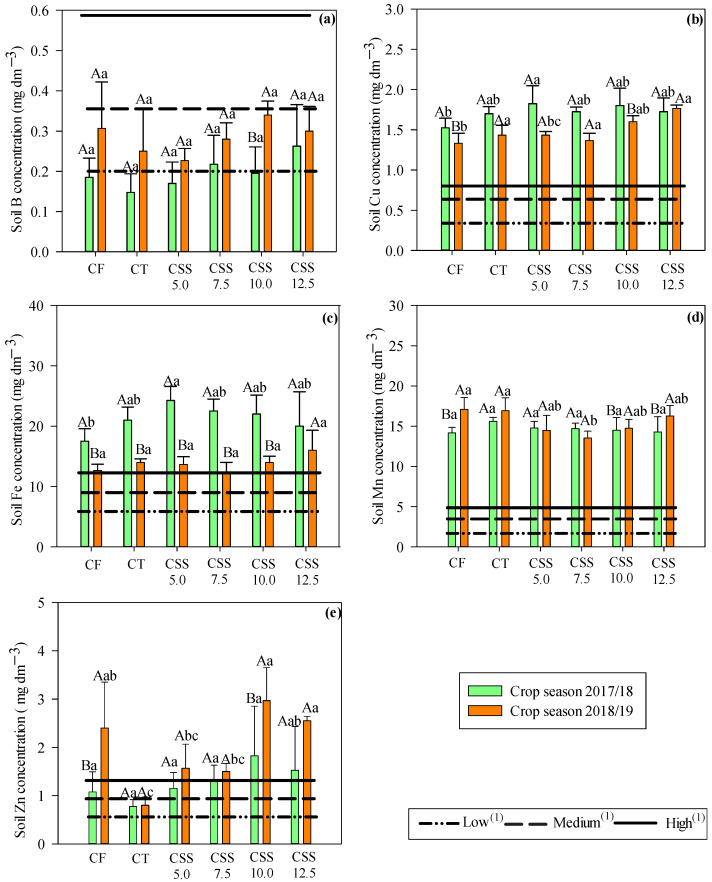
Effect of treatments (CF = conventional mineral fertilization; CT = control treatment (no CSS neither CF application); CSS = composted sewage sludge application at indicated rates in Mg ha^−1^, wet basis) on available soil B (**a**), Cu, (**b**), Fe (**c**), Mn (**d**), and Zn (**e**) concentrations (depth 0–0.20 m) after cowpea cultivation common in two agricultural years. Significant differences (*p* ≤ 0.05; Tukey test) between treatments (lowercase letters) or agricultural year (uppercase letters) are indicated by different letters. Error bars indicate standard deviation of means (*n* = 4). ^(1)^ Interpretation limits of micronutrient availability in tropical and subtropical soils for annual crops [11].

**Figure 2 plants-12-02153-f002:**
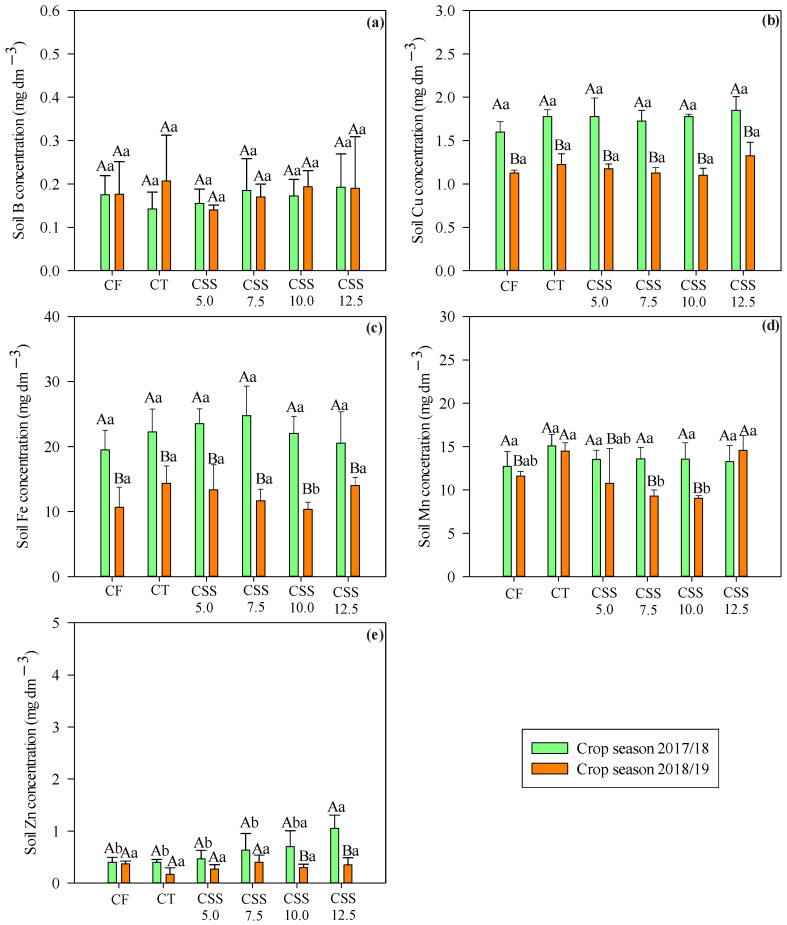
Effect of treatments (legend as in Figure 1) on soil available B (**a**), Cu, (**b**), Fe (**c**), Mn (**d**), and Zn (**e**) concentrations (depth 0.20–0.40 m) after cultivation of common bean in two agricultural years. Significant differences (*p* ≤ 0.05; Tukey test) between treatments (lowercase letters) or agricultural years (uppercase letters) are indicated by different letters. Error bars indicate standard deviation of means (*n* = 4).

**Figure 3 plants-12-02153-f003:**
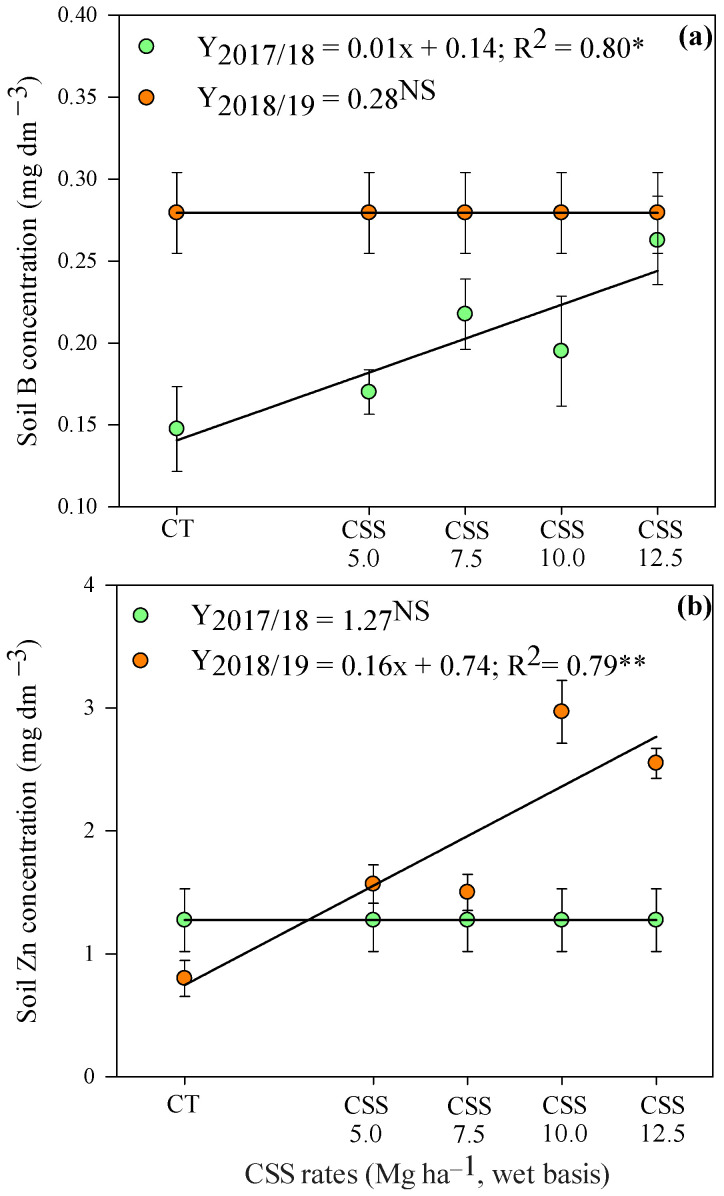
Availability of B (**a**) and Zn (**b**) in the soil at the 0 to 0.2 m depth layer after common bean cultivation in response to residual applications of composted sewage sludge (CSS) rates. **, * and ^NS^—Significant at 1 and 5% and not significant, respectively. Error bars indicate the standard deviation of the mean (*n* = 4). Legend as in Figure 1.

**Figure 4 plants-12-02153-f004:**
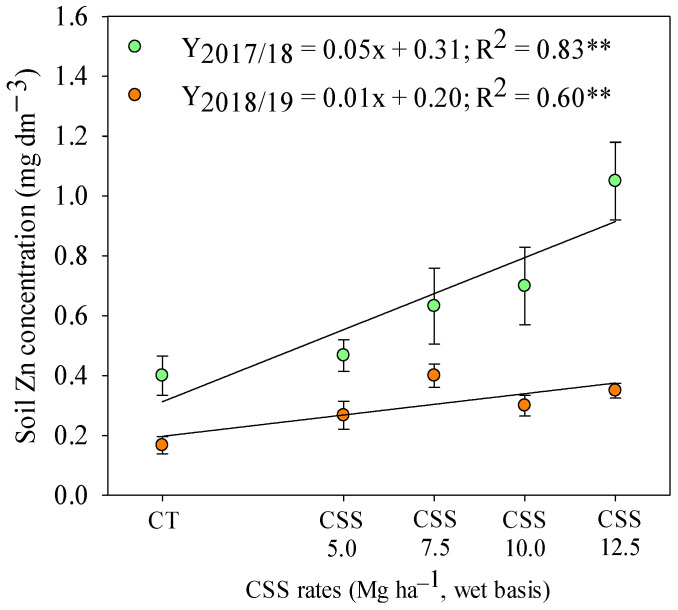
Soil Zn availability at the 0.2–0.4 m depth layer after common bean cultivation in response to residual application of composted sewage sludge (CSS) rates. **—Significant at 1%. Legend as in Figure 1.

**Figure 5 plants-12-02153-f005:**
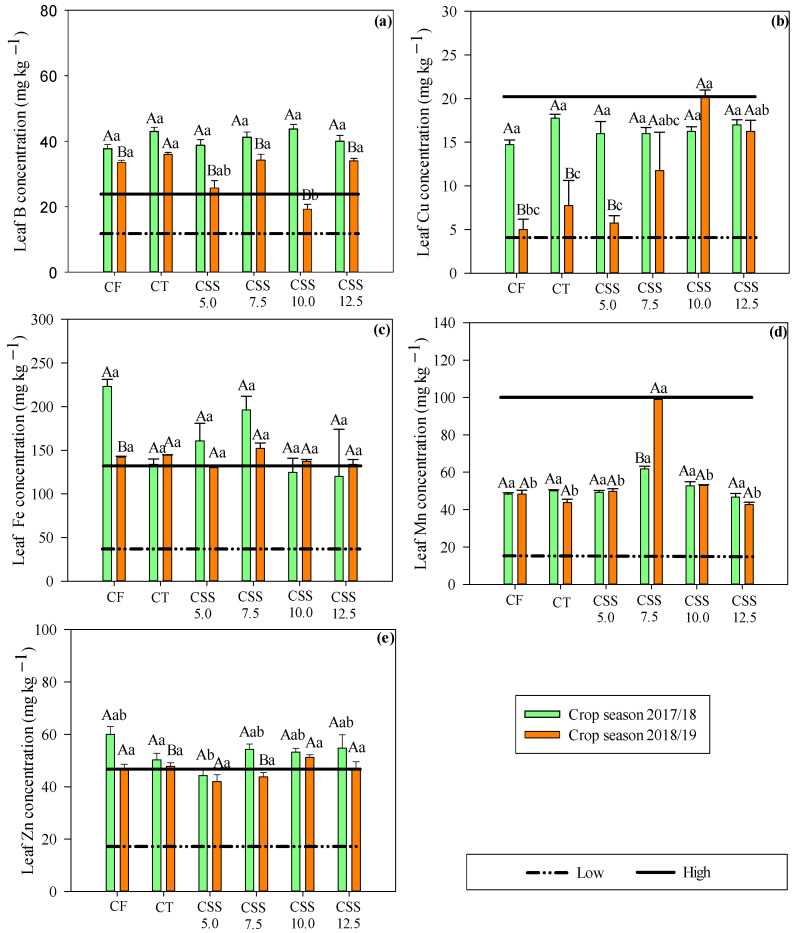
Micronutrient concentrations—B (**a**), Cu, (**b**), Fe (**c**), Mn (**d**), and Zn (**e**)—in common bean leaves in response to the effect of treatments (legend as in Figure 1) and of the evaluated agricultural years. Significant differences (*p* ≤ 0.05; Tukey test) between treatments (lowercase letters) or agricultural year (uppercase letters) are indicated by different letters. Error bars indicate the standard deviation of the mean (*n* = 4). The horizontal lines on graph bars represent range of interpretation of micronutrient concentrations for common bean crop as described by Ambrosano et al. [28].

**Figure 6 plants-12-02153-f006:**
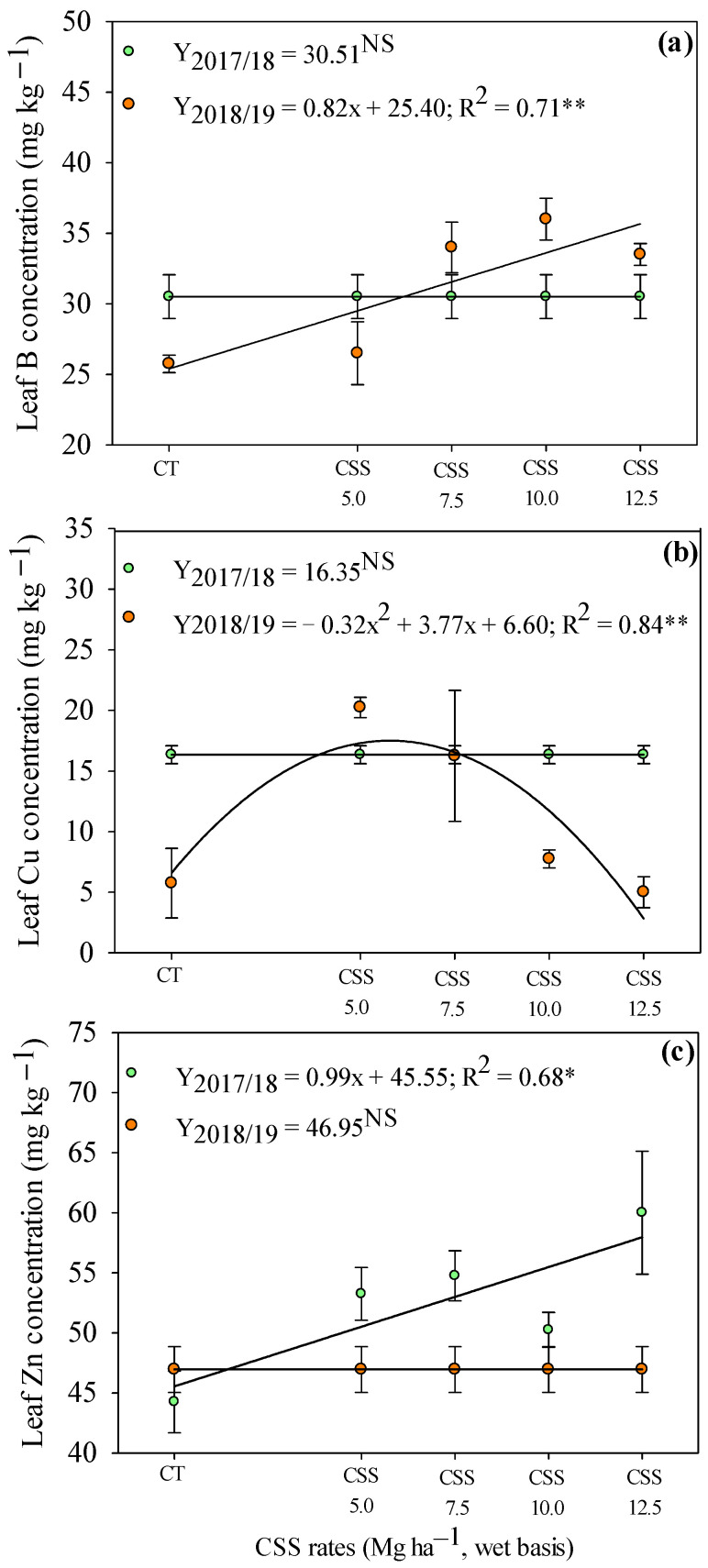
Concentrations of B (**a**), Cu (**b**), and Zn (**c**) in common bean leaf in response to the residual of the application of rates composed of sewage sludge (CSS). **, * and ^NS^—significant at 1 and 5% and not significant, respectively. Error bars indicate the standard deviation of mean (*n* = 4). Legend as in Figure 1.

**Figure 7 plants-12-02153-f007:**
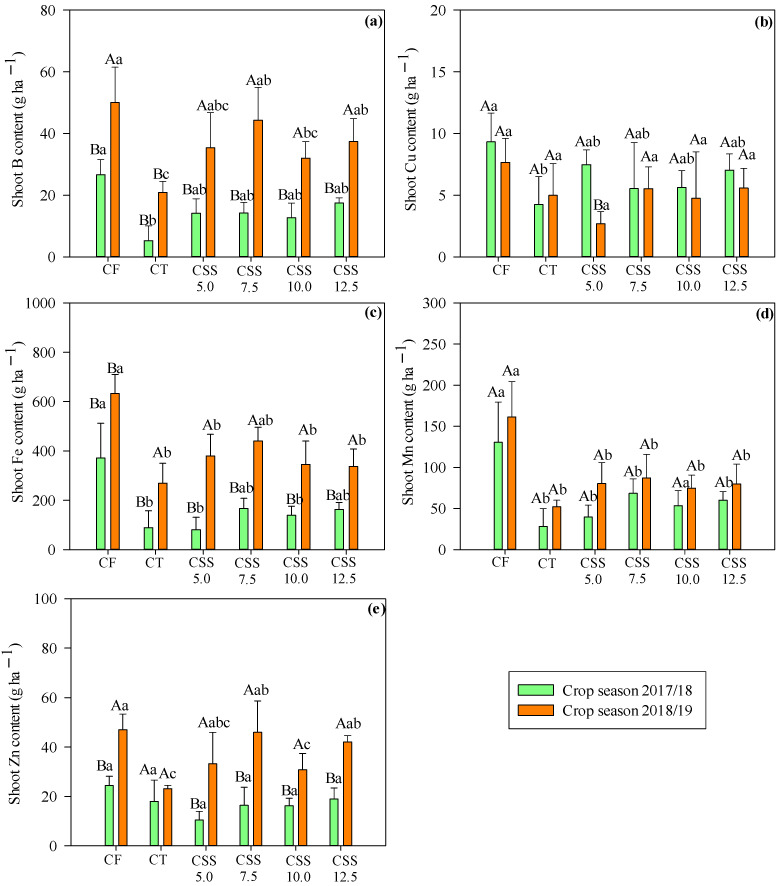
Micronutrient extraction–B (**a**), Cu, (**b**), Fe (**c**), Mn (**d**), and Zn (**e**)–by the common bean crop in response to the effect of treatments (legend as in Figure 1) and of the evaluated agricultural years. Means followed by the same letter (lowercase for treatments and uppercase for agricultural years) do not differ from each other via Tukey’s test at 5% probability. Error bars indicate the standard deviation of the mean (*n* = 4).

**Figure 8 plants-12-02153-f008:**
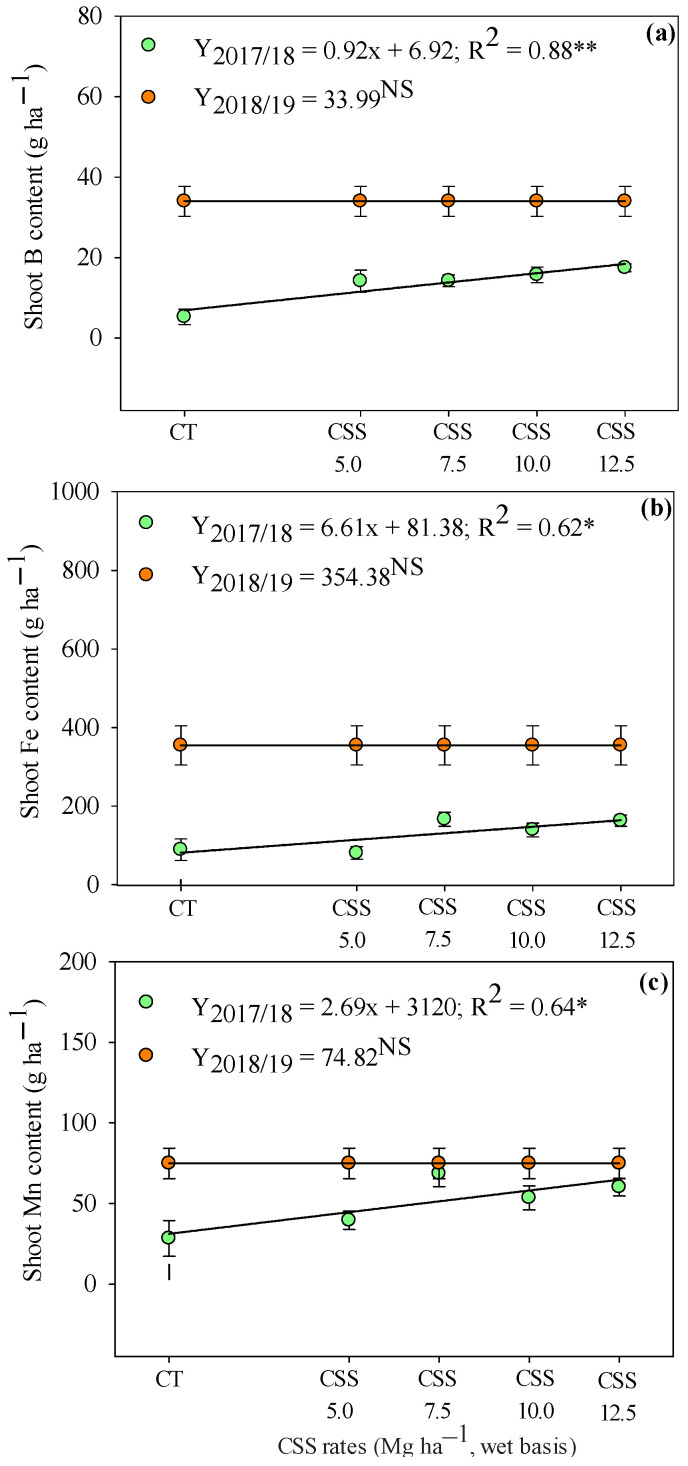
Extraction of B (**a**), Fe (**b**), and Mn (**c**) by common bean crop in response to residual application of sewage sludge compost (CSS) rates. **, * and ^NS^—significant at 1 and 5% and not significant, respectively. Error bars indicate the standard deviation of the mean (*n* = 4). Legend as in Figure 1.

**Figure 9 plants-12-02153-f009:**
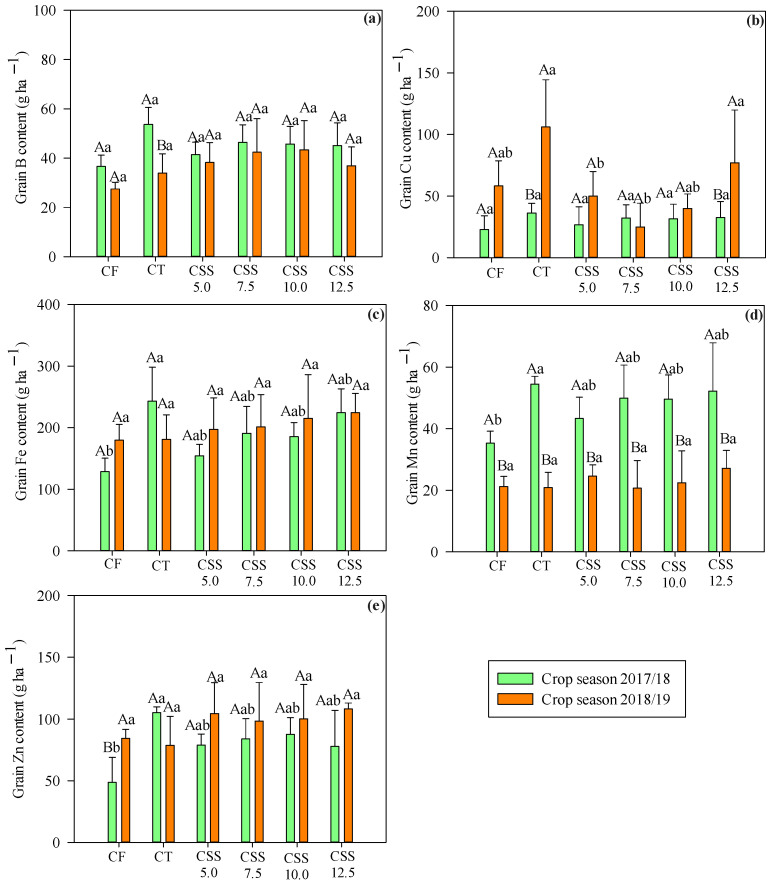
Micronutrients exported–B (**a**), Cu, (**b**), Fe (**c**), Mn (**d**), and Zn (**e**)–by winter common bean grains in response to the effect of treatments (legend as in Figure 1) and of the evaluated agricultural years. Means followed by the same letter (lowercase for treatments and uppercase for agricultural years) do not differ from each other in Tukey’s test at 5% probability. Error bars indicate standard deviation of the mean (*n* = 4).

**Figure 10 plants-12-02153-f010:**
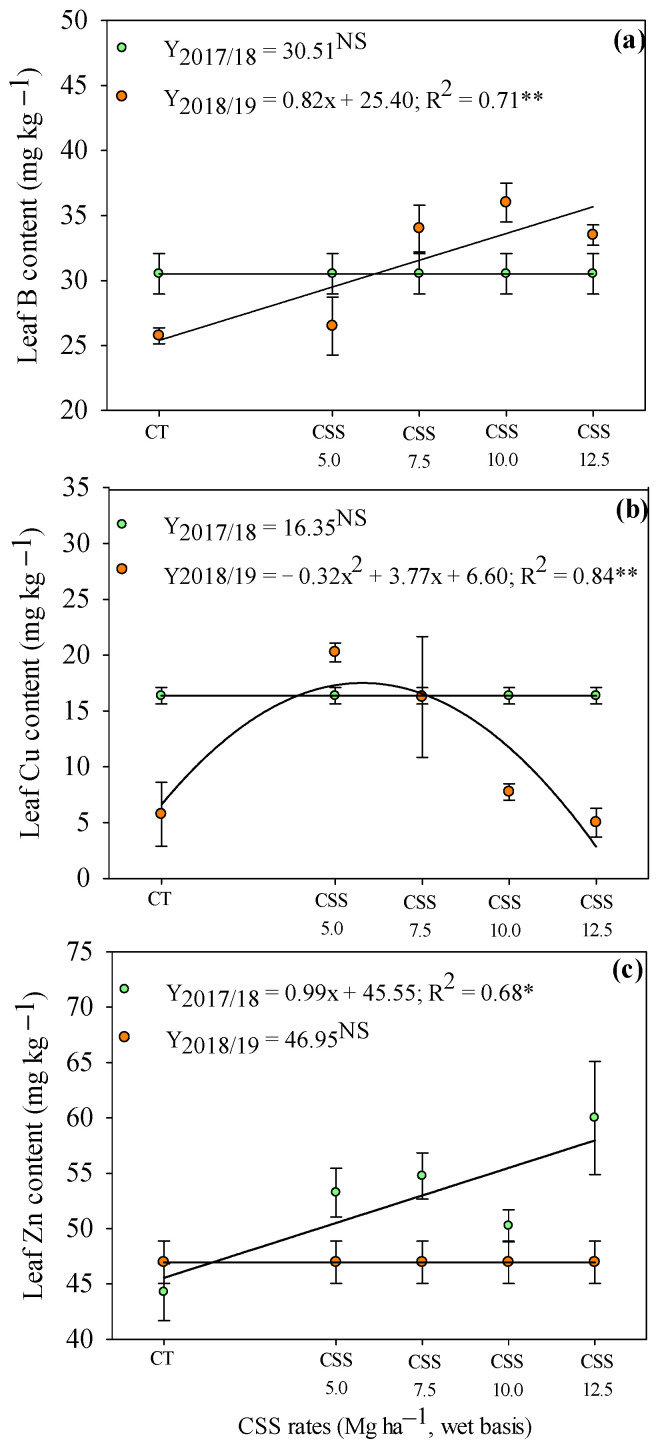
Export of Fe (**a**), Mn (**b**), and Zn (**c**) in common bean grains in response to the residual application of sewage sludge compost (CLE) rates. **, * and ^NS^—significant at 1 and 5% and not significant, respectively. Error bars indicate the standard deviation of the mean (*n* = 4). Legend as in Figure 1.

**Figure 11 plants-12-02153-f011:**
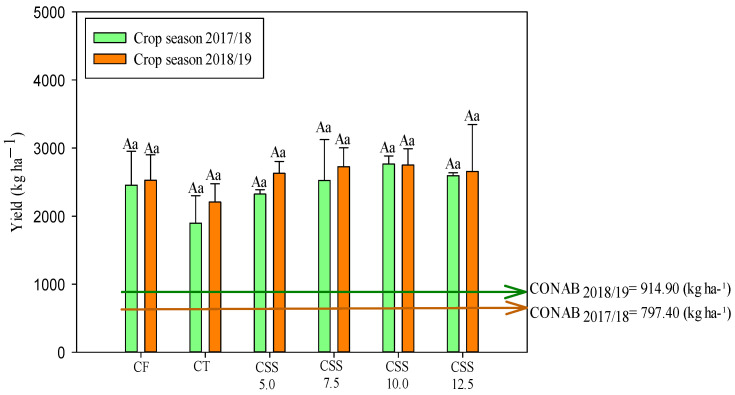
Common bean productivity in response to the effect of applied treatments and years of cultivation of the common bean crop in response to the effect of treatments (legend as in Figure 1) and the evaluated agricultural years. Means followed by the same letter (lowercase for treatments and uppercase for agricultural years) do not differ from each other in Tukey’s test at 5% probability. Error bars indicate the standard deviation of the mean (*n* = 4). The orange and green arrows indicate the national productivity of the second-crop bean crop grown in the 2017/18 [29] and 2018/19 [29] agricultural years, respectively.

**Figure 12 plants-12-02153-f012:**
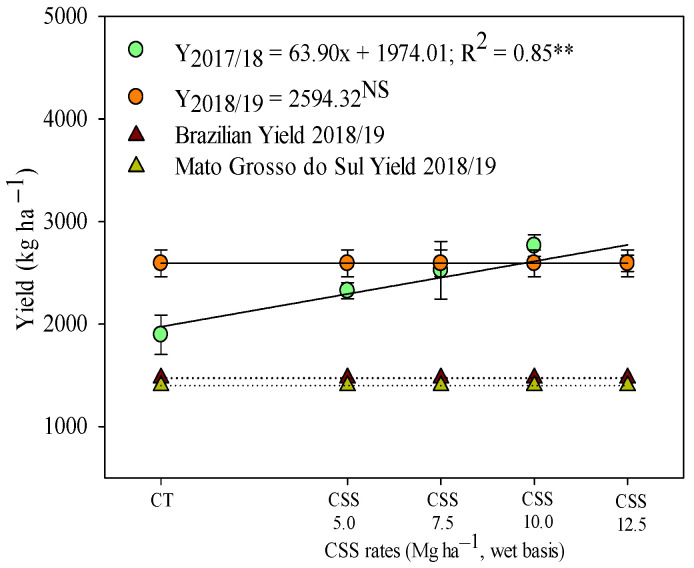
Yield of common bean in response to residual application of composted sewage sludge (CSS) rates. ** and NS—significant at 1% and not significant, respectively. Productivity averages of color bean second crop [29]. Legend as in Figure 1.

**Figure 13 plants-12-02153-f013:**
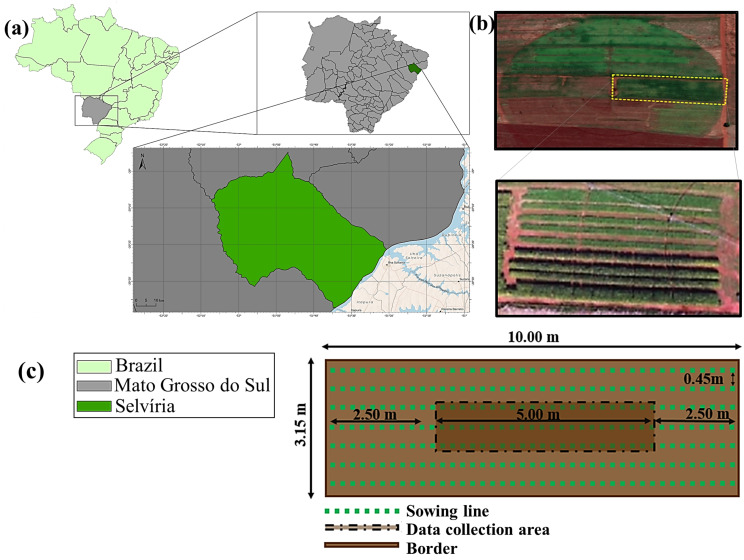
Experimental area located in the municipality of Selvíria at: 20°20′35″ S, 51° 24′04″ W; 358 m altitude; Mato Grosso do Sul State—MS, Brazil (**a**); aerial view of the entire experimental area and plots (**b**); schematic representation of a single plot with the individuation of the “useful area” for collecting soil and plant data (**c**).

**Figure 14 plants-12-02153-f014:**
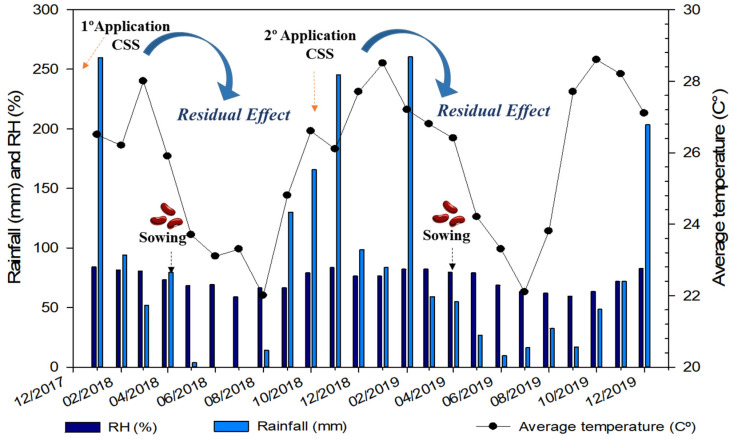
Rainfall, relative humidity (RH), and average temperatures during common bean cultivation in the 2017/18 and 2018/19 growing seasons. Data were collected from the meteorological station of the Faculty of Engineering of Sao Paulo State University, Ilha Solteira, Brazil.

**Table 1 plants-12-02153-t001:** Characterization of some chemical ^(1)^ and physical ^(2)^ attributes of the soil before the installation of the experiment (mean ± standard deviation; *n* = 3).

Attributes	Unit	Depth (m)	
		0–0.20	0.20–0.40
pH (CaCl_2_)	-	4.5 ± 0.06	4.7 ± 0.06
Organic matter	g dm^−3^	19 ± 1.16	14 ± 0.58
Phosphor	mg dm^−3^	16 ± 0.58	9 ± 0.00
Potassium	mmol_c_ dm^−3^	1.7 ± 0.17	0.7 ± 0.15
Calcium	mmol_c_ dm^−3^	13 ± 0.58	11 ± 0.58
Magnesium	mmol_c_ dm^−3^	12 ± 1.00	10 ± 0.00
Aluminum	mmol_c_ dm^−3^	4 ± 0.00	2 ± 0.58
H + Al	mmol_c_ dm^−3^	37 ± 2.31	32 ± 1.73
SB	mmol_c_ dm^−3^	27.0 ± 1.69	22.1 ± 0.72
S-SO4	mg dm^−3^	15 ± 0.58	8 ± 0.58
CEC	mmolc dm^−3^	63.7 ± 0.86	54.1 ± 2.45
BS	%	42 ± 3.21	41 ± 0.58
Boron	mg dm^−3^	0.22 ± 0.04	0.15 ± 0.02
Copper (DTPA)	mg dm^−3^	1.8 ± 0.05	1.4 ± 0.10
Iron (DTPA)	mg dm^−3^	15 ± 0.58	8 ± 0.58
Manganese (DTPA)	mg dm^−3^	18.8 ± 0.59	7.3 ± 0.72
Zinc (DTPA)	mg dm^−3^	0.6 ± 0.06	0.2 ± 0.00
Sand (>0.05 mm)	g kg^−1^	553 ± 12.86	
Silt (>0.002 and <0.05 mm)	g kg^−1^	81 ± 3.21	
Clay (<0.002 mm)	g kg^−1^	372 ± 19.05	
Texture	-	clayey	

^(1)^ Raij et al. [31]. ^(2)^ Teixeira et al. [32]. CEC = Cation exchange capacity. SB = Sum of bases. BS = Base saturation.

**Table 2 plants-12-02153-t002:** Chemical and microbiological composition of composted sewage sludge samples (mean ± standard deviation; *n* = 3).

	Unit	Values		Limits ^1^
Chemical Features		2017/18	2018/19	
pH (CaCl_2_)	-	7.0 ± 0.1	7.3 ± 0.1	-
Moisture (60–65 °C)	%	41.0 ± 0.3	34.4 ± 0.5	-
Total moisture	%	45.5 ± 0.2	35.8 ± 0.6	-
Total OM	g kg^−1^	308.7 ± 10.0	255.0 ± 7.4	-
CEC	mmol_c_ kg^−1^	520.0 ± 20.0	--	-
C/N	-	12.0 ± 0.8	9.0 ± 0.6	-
Total N	g kg^−1^	13.9 ± 0.3	15.3 ± 1.5	-
Total P	g kg^−1^	12.3 ± 1.4	14.1 ± 0.0	-
Total S	g kg^−1^	4.8 ± 0.3	8.4 ± 1.4	-
Na	mg kg^−1^	3930.0 ± 32.0	3915.0 ±32.0	-
K	g kg^−1^	6.0 ± 2.2	8.2 ± 0.4	-
Ca	g kg^−1^	19.4 ± 4.4	31.1 ± 1.1	-
Mg	g kg^−1^	5.2 ± 0.5	9.9 ± 0.2	-
As	mg kg^−1^	3.2 ± 1.8	-	20.0
B	mg kg^−1^	94.0 ± 4.5	94.0 ± 4.6	NR
Cd	mg kg^−1^	1.0 ± 0.1	-	3.0
Cu	mg kg^−1^	237.0 ± 16.5	191.2 ± 5.8	NR
Pb	mg kg^−1^	18.1 ± 1.6	-	150.0
Cr	mg kg^−1^	54.3 ± 1.8	-	2.0
Fe	mg kg^−1^	16,400 ± 1300	14,708 ± 249	NR
Mn	mg kg^−1^	246.0 ± 37.0	310.0 ± 15.0	NR
Hg	mg kg^−1^	0.22 ± 0.1	-	1.0
Mo	mg kg^−1^	5.26 ± 0.2	-	NR
Ni	mg kg^−1^	26.5 ± 0.5	-	70.0
Zn	mg kg^−1^	456 ± 8	684 ± 7	NR
biological analysis				
Salmonella sp.	MPN 10 g^−1^	Absent		Absent
fecal coliform	MPN g^−1^	0		<103 MPN g^−1^ on dry weight
Viable helminth eggs	Eggs g^−1^ on dry weight	0.12		<10 Eggs g^−1^ on dry weight

^1^ Limits to organic fertilizers use established by the Ministry of Agriculture, Livestock and Food Supply in Brazil [33]. PTE = potentially toxic elements; NR = not ruled; MPN = most probable number.

## Data Availability

The information and database for this research are currently not on a platform or website. They can be provided by the corresponding author.

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
