# Peer review of "Common Bean Productivity and Micronutrients in the Soil–Plant System under Residual Applications of Composted Sewage Sludge"

_plants, 2023, doi:10.3390/plants12112153_

Round 1
Reviewer 1 Report
Dear Authors,
Based on my previous expertise, your manuscript titled "Common bean productivity and micronutrients in the soil-plant system under residual applications of composted sewage sludge" is of high quality, and contains important results for plant scientists.
I suggest to read through the entire manuscript, the English style and grammar need some corrections.
Anyway, I suggest to maintain the structure and content of the manuscript as it is now, it is clear and understandable, with quality discussion and acceptable conclusions.
Kind regards
English language is fine, minor grammatical mistakes can be found, correction is needed.
Reviewer 2 Report
See attached file
MATERIALS AND METHODS
Line 443 - The authors should include table 1 and a comment on the chemical-physical characteristics of the soil because some of them, for example the pH, the cation exchange capacity, the organic carbon content, influence the dynamics of micronutrients and heavy metals in the soil system- plant.
Line 454- In Figure 14 check and replace HR with RH.
Line 473 - The authors should insert table 2 in the text of the paper. Some characteristics of the compost, as well as for the soil, can affect the mobility of nutrients when compost is applied to the soil.
Line 463- Authors should assign a acronym to each treatment (CSS rates and Control), not just conventional fertilization. These acronyms must be reported in the text and in all the figures in order to allow each measured value to be associated with the relative treatment. Otherwise the results and figures are difficult to read.
527 “Nutritional parameters of bean culture” The authors should better detail the preparatory phase of the plant samples because in materials and methods they write only leaves. In the results they also write about grains and shoots. Also, how did they determine the crop productivity?
Line 536-540 Authors should clarify in paper what they mean by export and how they calculated it.
Do they mean the uptake of micronutrients which is obtained by multiplying the concentrations in plant per dry plant biomass?
RESULTS
Line 106 - The X labels of the different treatments must be reported in all the graphs under the x axis (fig.1,2, 5-10). In this version of the paper, reading the results and comparing them with the graphs is neither simple nor immediate.
Line 217- “Extraction and export of micronutrients” in shoots, crop, grains? Authors should specify for consistency with discussion of data.
What do the authors mean by extraction and export? Authors should clarify in paper. Regarding the micronutrients export, how was it calculated?
Do they mean the uptake of micronutrients which is obtained by multiplying the concentrations in plant per dry plant biomass?
Line 218 - shoots? crop? Authors should be consistent with the terminology used in all sections of the text and in figure 7, 8 captions.
Line 262 - export: see previous comment line 217
“micronutrients carried out by bean plant” plant or grains? (see fig. 9 caption)
Line 292 specify grains export
DICUSSION
Line 329 The authors did not present data on soil organic matter. Did they analyze the soil before and at the end of the first and second year of experimentation? They should present the data.
Line 336 - The behavior of Cu and Zn, both in total and available form, depends on the type of bonds they form with the different fractions of organic matter in the soil. The authors should present at least the physico-chemical characterization of the soil in order to broaden the discussion on the behavior of these two metals in the two years of experimentation.
Line 354 -What is the meaning of this sentence? It is an incomplete sentence
Line 359-365 - This sentence should be placed in the introduction.
Line 377 - The authors did not present Table 1 in the supplementaty files. However, table 1 with the characterization of the soil should be presented in the text.

Round 2
Reviewer 2 Report
The authors have made the suggested changes and improved the whole paper